# Therapeutic Lymphangiogenesis Is a Promising Strategy for Secondary Lymphedema

**DOI:** 10.3390/ijms24097774

**Published:** 2023-04-24

**Authors:** Yuuki Shimizu, Yiyang Che, Toyoaki Murohara

**Affiliations:** Department of Cardiology, Nagoya University Graduate School of Medicine, Nagoya 466-8550, Japan

**Keywords:** lymphedema, therapeutic lymphangiogenesis, translational research, review article

## Abstract

Secondary lymphedema is caused by lymphatic insufficiency (lymphatic drainage failure) following lymph node dissection during the surgical treatment or radiation therapy of breast or pelvic cancer. The clinical problems associated with lymphedema are reduced quality of life in terms of appearance and function, as well as the development of skin ulcers, recurrent pain, and infection. Currently, countermeasures against lymphedema are mainly physical therapy such as lymphatic massage, elastic stockings, and skin care, and there is no effective and fundamental treatment with a highly recommended grade. Therefore, there is a need for the development of a fundamental novel treatment for intractable lymphedema. Therapeutic lymphangiogenesis, which has been attracting attention in recent years, is a treatment concept that reconstructs the fragmented lymphatic network to recover lymphatic vessel function and is revolutionary to be a fundamental cure. This review focuses on the translational research of therapeutic lymphangiogenesis for lymphedema and outlines the current status and prospects in the development of therapeutic applications.

## 1. Introduction

Similar to blood vessels, lymphatic vessels are present in organs throughout the body and play a crucial role in draining interstitial fluid (lymph fluid) containing lipids, proteins, and immune cells from the interstitial spaces between cells. This fluid is then returned to the venous circulation via lymph nodes and the thoracic duct [1]. Thus, the lymphatic system, including lymphatic vessels, is the central organ that maintains homeostasis in the interstitium and has an important role as one of the immune surveillance systems [1]. 

Secondary lymphedema develops when lymph node dissection during surgery or radiotherapy for breast or pelvic cancer causes lymphatic dysfunction, resulting in the inadequate drainage of interstitial fluid in peripheral limbs. This disease poses significant clinical challenges to patients, affecting their appearance and quality of life (QOL) due to limb dysfunction, skin ulceration, recurrent pain, and the risk of infection (cellulitis) [2]. Although the number of patients with secondary lymphedema is expected to increase in the future due to advances in radical surgery and the extended life expectancy of cancer survivors, current treatments are mainly anti-symptomatic therapies such as lymphatic drainage massage, wearing elastic stockings, and skin care, and highly recommended radical treatments are extremely limited [2]. Therefore, there is an unmet medical need to develop new treatments for refractory lymphedema. Therapeutic lymphangiogenesis is a promising new strategy to reconstruct the fragmented lymphatic network and restore lymphatic function. 

This review aims to highlight the potential of therapeutic lymphangiogenesis, particularly in treating refractory lymphedema, and provide a future perspective on this promising therapeutic strategy.

## 2. Lymphatic Vessel

Lymphatic vessels, known as lymphatics, are thin tubes that collect and transport the lymphatic fluid through the lymphatic system (low-pressure, unidirectional flow) [3]. In contrast, during mammalian embryonic and adult growth, blood vessels supply oxygen and nutrients to all cells [4]. Despite the structural linkage and joint function of the lymphatic and blood-vascular systems, they are formed through different molecular processes. Vasculogenesis and angiogenesis, primarily driven by the vascular endothelial growth factor (VEGF), create the blood vessels [5]. On the other hand, lymphangiogenesis is activated by VEGF-C, which affects LECs that separate from venous endothelial cells at mid-gestation [6]. In addition, LECs exhibit specific markers such as lymphatic artery hyaluronan receptor-1 (LYVE-1), vascular endothelial growth factor receptor-3 (VEGFR-3), membrane glycoprotein podoplanin, and Prospero-related homeodomain transcription factor (Prox1) [7,8,9]. Recently, there have been significant breakthroughs in the study of lymphatic molecular markers, with a particular emphasis on the system’s distinctive cellular and functional characteristics and its function in pathology [10]. It has been well-documented that lymphatic vessels play critical roles in removing interstitial fluid produced by blood vessel filtrates and supporting tissue immunosurveillance while coordinating with other immune responses. The lymphatic system consists of a hierarchical network of vessels, nodes, and organs that work together to maintain fluid balance, filter foreign particles, and mount an immune response against infection. At the smallest scale, lymphatic capillaries are the smallest vessels in the network and are found in the interstitial spaces between cells. These capillaries have thin walls and are permeable, allowing fluids, proteins, and immune cells to enter them. They are also responsible for the uptake of fats from the digestive system. Lymphatic collecting vessels are larger than capillaries and have a similar structure to veins. They are lined with endothelial cells and have smooth muscle cells in their walls that help to propel lymph fluid through the vessels. Lymphatic vessels are found throughout the body and converge to form larger vessels that eventually empty into the bloodstream. Specific body systems, such as the intestinal lymphatic vessels, aid in maintaining gut homeostasis and immunity, and the lymphatic vasculature is essential for central nervous system (CNS) function, facilitating cerebrospinal fluid (CSF) clearance [1]. Studies have also shown that lymphatic mimicry is a mechanism of vascular adaptation to organ-specific functions [11]. However, lymphatic system malfunction can contribute to the pathogenesis of diseases involving inflammatory and immunological reactions and the dissemination of tumor cells. Insufficient lymphatic function can lead to the accumulation of connective tissue and obesity, reduced immune reactions, lymph buildup in tissues, and lymphedema [12]. 

## 3. Lymphedema

### 3.1. Causes

A chronic pathologic state of the lymphatic system known as lymphedema is characterized by the accumulation of protein-rich fluid in the interstices and the subsequent development of inflammation, adipose tissue hypertrophy, and fibrosis. Moreover, reduced movement and function, as well as disfigurement, can result from the swelling and following induration of the afflicted area [13]. Lymphedema could be classified by the etiology of the disease state of patients and is therefore categorized into primary lymphedema and secondary lymphedema. 

Primary lymphedema is caused by abnormalities in lymphatic system development that leave the lymphatics underdeveloped or malfunctioning. Symptoms of these diseases can occur at birth or typically appear during adolescence or adulthood. Secondary lymphedema, which is more frequent, is caused by infections, injuries, or obstructions to the lymphatic system [13]. Primary lymphedema is associated with heritable chromosomal abnormalities and affects how the lymph vessels form [14]. It can be classified based on the age of the patient’s emergence of symptoms. It can be clinically classified as “congenital lymphedema”, which appears at birth or within the first 2 years of age; “lymphedema precox”, which manifests from 2 to 35 years of age; or “lymphedema tarda”, which is onset after 35 years of age [15]. The majority of incidences of congenital lymphedema are triggered by mutations in the FMS-like tyrosine kinase 4 (FLT4) gene [16]. FLT4 gene encodes vascular endothelial growth factor receptor 3 (VEGFR-3), which controls lymphatic system formation and maintenance. Some occurrences of lymphedema precox and tarda have been related to mutations in the forkhead box C2 (FOXC2) gene, which is required for lymphatic valve development [17]. Mutations in these genes create lymphatic system anomalies that prevent fluid from draining effectively, resulting in lymphedema.

Secondary lymphedema is subdivided by blockade at the lymph node level and disruption or obliteration of lymphatic channels. Furthermore, the causes of secondary lymphedema could be a range of infections, tumors, trauma and tissue damage, inflammation, obesity, and venous diseases [18]. Infections caused for secondary lymphedema include filariasis, cellulitis/erysipelas, tuberculous lymphadenitis, and lymphogranuloma venereum. However, cancer-related treatment is the most common cause of secondary lymphedema in the United States and developed countries [18]. Malignant-disease-caused secondary lymphedema includes the excision of the lymph node basin, radiation treatment, lymphatic metastasis, and subsequent lymph channel functional impairment. Trauma and tissue damage often result from surgery or injury damage to lymphatic vessels. Moreover, inflammation causes a range of diseases from rheumatoid arthritis to dermatoses, including epidermal dermatoses, psoriasis, and sarcoidosis. The National Lymphedema Network defines people at risk for lymphedema as those who have not yet exhibited signs and symptoms associated with a lymphedema diagnosis but have a confirmed lymphatic system insufficiency such as older age, excess weight or obesity, and rheumatoid or psoriatic arthritis. In addition, those with lymph nodes removed or radiation therapy as part of cancer treatment or those with genetic lymphedema in their family may also be at risk [19]. 

### 3.2. Epidemiology

Primary lymphedema is an uncommon disease, and the estimation of the prevalence of heritable causes of lymphedema is challenging to obtain and varies greatly. Primary lymphedema is believed to impact nearly 1 in every 6000 to 10,000 live births, with adolescence being related to the appearance of early onset edema [20]. Primary lymphedema is most common in females and often affects females 2- to 10-fold more frequently than males [21]. In primary lymphedema, lymphedema precox is the most common type [22], while lymphedema tarda is relatively uncommon and could account for fewer than 10% of cases of primary lymphedema [23]. 

Compared with primary lymphedema, secondary lymphedema is much more common and often occurs later in life [24], and it affects nearly 1 in 1000 Americans. In addition, the epidemiology of secondary lymphedema is widely examined in oncologic patients. The incidence of unilateral limb lymphedema after breast cancer therapy varied from 8.4 to 21.4%, according to a 2013 comprehensive review and meta-analysis [25]. Another systematic analysis assessed the prevalence of secondary lymphedema owing to non-specific cancer to be 2.05–3.99:1000 in lymphedema specialty clinics in the United Kingdom (UK) (n = 11,555) [26]. It is estimated that approximately 3500 patients per year develop lymphedema of the lower and upper extremities as a result of treatment for uterine or breast cancer in Japan [27]. In the reported research, the rates of upper-extremity lymphedema (UEL) after axillary lymph node dissection (ALND) for melanoma ranged from 4.4% to 14.6% and from 4.1% to 21.4% after ALND for breast cancer [28]. In addition, secondary lymphedema of the lower limb is one of the most severe complications linked with diagnosing and treating gynecologic cancer. It is widely assumed that 20% to 60% of those suffering from gynecologic cancer will experience lymphedema [29,30]. Secondary lymphedema is common among individuals with prior head and neck surgery or radiation treatment. After treatment for head and neck cancers, up to 75% of people will develop several signs and symptoms of lymphedema [31,32,33]. With the treatment of genitourinary malignancies, lymphedema happened in 4% of prostate cancer patients, 16% of bladder cancer patients, and 21% of penile cancer patients [34,35]. 

### 3.3. Symptoms and Complications

The major symptom of primary lymphedema is edema in various areas of the body caused by lymph accumulation in the soft layers of tissue beneath the epidermis. The degree of edema can increase in all forms of primary lymphedema. Edema is often encountered in one or both legs but can also be found in the torso, face, genitalia, and limbs [36]. Swelling can cause tightness, pain, and strange tingling feelings (paresthesias) in the afflicted regions of some people. The afflicted area’s skin may become unusually dry, thickened, or scaly (hyperkeratosis), resulting in a “woody” appearance [37]. Leg edema is the most common symptom of congenital lymphedema, but the genitals may also be impacted in some individuals. In addition to the lymphedema of the legs, other regions of the body, such as the arms, face, and larynx, may also be impacted in lymphedema praecox [38]. Some people may acquire yellow cuticles [39]. The edema of lymphedema tarda mainly affects the legs, but it can also affect the arms and other regions. In women, the lower limbs are the most impacted [40]. As for secondary lymphedema, patients are suffering from swelling, paraesthesia, skin changes such as tight and shiny skin, reduced range of movement in affected limb joints, heaviness, functional restriction, and pain caused by inflammation, infection muscle strain, nerve compression, or reduced function (CREST guidelines 2008). Early signs and symptoms of lymphedema should also be looked out for in case of noticing lymphedema. They appear as tighter clothing or jewelry, a feeling of heaviness and tightness, aching, and observable swelling. Additional complications sometimes associated with lymphedema include skin infections such as cellulitis, a common infection in lymphedema [41,42]. In some individuals with lymphedema, vein thrombosis could also happen in the upper or lower limb, possibly related to sudden swelling [43]. In addition, the lymph fluid could drain through small breaks in the skin or cause blistering in patients with severe lymphedema [44]. The skin of the afflicted leg can thicken and harden in some individuals with extreme lymphedema, resembling elephant skin [45]. 

### 3.4. Conventional Treatment

Treatment for lymphedema aims to alleviate the symptoms, avoid progression, and lower the risk of skin infection. Here, we list several types of therapy that currently exist to treat lymphedema.

#### 3.4.1. Skin Care

Skin care is emphasized in lymphedema treatments. Skin care aims to reduce dermal colonization by bacteria and fungus, eradicate bacterial and fungal overgrowth in skin crevices, and hydrate the skin to regulate dryness and eliminate cracking. Keeping the skin clean is essential because it can help avoid skin issues such as redness, itching, or developing a rash or irritation [46]. However, skin care treatment is not effective for all types and stages of lymphedema and is not a fundamental therapy for lymphedema.

#### 3.4.2. Compression Garments

Compression therapy with fitted garments is an essential aspect of therapy, with most studies showing volume decreases compared to pre-treatment values, although stated mean volume reductions differ significantly [47]. Compression garments prevent lymphedema from worsening. They can help with minor edema. In the case of serious lymphedema, multi-layered bandaging is advised. The pressure applied to the organs by the clothing varies. In the early stages, compression garments are not necessary since lymphedema is mild. In advanced stages, the skin becomes thickened and fibrotic, making it difficult for compression garments to effectively compress the affected area. In addition, compression garments may be uncomfortable or difficult to wear for some individuals.

#### 3.4.3. Lymphatic Massage

Lymphatic massage aims to shift fluid from the swollen region to a location where the lymphatic system is functioning regularly [48]. Manual lymphatic drainage (MLD) is a component of complete decongestive therapy (CDT) recommended after surgery to reduce the risk of lymphedema. MLD involves a gentle massage technique that follows the lymphatic pathways and can be performed daily or up to three times a week for three or more weeks. Maintenance sessions can be repeated every three months to a year to prevent the development of lymphedema. Studies have demonstrated the efficacy of MLD in reducing the onset of lymphedema following surgery. Simple lymphatic drainage (SLD) is a self-administered form of MLD that patients and carers can learn and use. SLD has no definitive techniques, but it is identical to MLD in that it is performed for 10–20 min every day [49]. Lymphatic massage can provide temporary relief from lymphedema symptoms, but the effect may only last for a short time. Therefore, it is typically used as a complementary therapy alongside other treatments for lymphedema.

#### 3.4.4. Multi-Layer Inelastic Lymphedema Bandaging

The standard intensive therapy for severe lymphedema is multilayer inelastic lymphedema bandaging (MLLB) administered at a pressure of >45 mmHg. MLLB is made up of several layers, as well as individual components with varying elastic characteristics. The combination of these layers causes the finished bandage’s elastic characteristic to alter, making it more inelastic [50]. It is typically used with other complicated lymphedema therapy (CLT) treatments to accomplish initial limb volume reduction. Compression hosiery is suggested for maintenance treatment once substantial limb volume reduction has been accomplished [51]. Multi-layer inelastic bandaging can restrict mobility and make it difficult to perform daily activities. This can affect the quality of life and may discourage some patients from using this treatment.

#### 3.4.5. Intermittent Pneumatic Compression

Intermittent pneumatic compression (IPC) takes over the function of destroyed lymphatics by compressing edema tissue fluid to normal lymphatic drainage areas [52]. It showed a long-term decrease in limb circumference and improved tissue elasticity. There were no complications, such as a thigh ring or chronic genital edema [53]. Patients with lower limb lymphedema can be successfully prescribed long-term, high-pressure IPC, extended inflation-timed treatment [54]. Intermittent pneumatic compression can provide temporary relief from lymphedema symptoms, but the effect may only last for a short time.

#### 3.4.6. Exercise

Recent studies suggest that lymphedema patients may benefit from exercise. Exercise, according to research, can help the range of motion and muscle of the affected extremities, as well as general fitness and functional quality of life, and can be performed without exacerbating lymphedema symptoms [55]. Aerobic activity could raise intra-abdominal pressure, which aids in thoracic duct circulation [43]. Combinations of flexibility, resistance, and aerobic exercise may help to manage lymphedema [56]. However, exercise is limited in the effect on relief from lymphedema and is not a fundamental therapy.

#### 3.4.7. Psychosocial Support

Lymphedema has a negative psychosocial effect on those who suffer from it. Psychosocial support is an essential component of lymphedema therapy. It can significantly impact the result by improving concordance, promoting self-management, and optimizing the quality of life. Interventions address age-related requirements, physical image, and social support [57,58]. However, psychosocial support care is limited in the effect on relief from lymphedema and is not a fundamental therapy.

#### 3.4.8. Palliative Care

Palliative rehabilitation’s primary objective is to establish therapy goals that enable a patient to keep or enhance functions while delaying disease progression for as much as possible. Palliative lymphedema evaluation seeks to understand the patient’s primary concerns, objectives, and priorities, as well as to assist the physician in understanding the fundamental cause and processes of the swelling, as well as how rapidly it is advancing [59]. Patients with lymphedema may be unable to endure a complete assessment and treatment program, necessitating a palliative strategy in which assessment methods are changed and particular therapies are chosen to alleviate specific symptoms. However, palliative care is limited in the effect on relief from lymphedema and is not a fundamental therapy.

#### 3.4.9. Surgery

The surgical therapy of lymphedema treatment could be divided into several categories: vascularized lymph node transfer (VLNT), lymphatic venous anastomosis (LVA), lymphaticolymphatic bypass, and suction-assisted protein lipectomy (SAPL) [60]. The VLNT technique includes microvascular anastomosing functional lymph nodes into an extremity to reestablish physiologic lymphatic function [61]. In most patients, LVA showed a significant decrease in limb volume and relief in subjective lymphedema results [62]. The lymphovenous bypass technique includes identifying obstructed lymphatic vessels and bypassing them into nearby venules [61]. LVA is a technically challenging procedure that requires highly skilled surgeons. The success of the procedure depends on the surgeon’s ability to identify and connect the appropriate lymphatic vessels to veins. Accordingly, the outcomes of LVA can vary depending on several factors, including the severity of lymphedema, the location of the lymphatic vessels, and the skill of the surgeon. SAPL has been shown to securely and efficiently decrease the solid component of swelling in chronic lymphedema. In addition, SAPL and VLNT can be joined to produce the best results in individuals with chronic lymphedema [63]. A limitation of VLNT is the risk of donor-site lymphedema.

#### 3.4.10. Other Treatments

Other treatments for lymphedema include drug treatment [64], hyperbaric oxygen [65], and laser therapy [66]. However, there is little evidence of drug treatment, hyperbaric oxygen, and laser therapy in the treatment of lymphedema. More research with bigger sample sizes, better methodological quality, and standardized and repeatable outcome metrics are needed to bring those therapies from the bench to the bedside.

## 4. Animal Lymphedema Model

Establishing suitable lymphedema models is critical for improving our knowledge of the disease’s molecular biology and pathogenesis and testing new treatments such as gene therapy or cell therapy (Figure 1). Here, we list several animal models of lymphedema created to investigate pathophysiology and therapy for lymphedema.

### 4.1. Primary Lymphedema Animal Model

As previously reported, Geng et al. used ProxTom, *Prox1*+/GFPCre (*Prox1*+/−), *Foxc2*+/−, *Gata2*+/−, Tg*VE*, *Gata2*f/f, and *Cx3*7+/− mice as a primary lymphedema model [67]. They demonstrated that LVV morphogenesis is disrupted in four different mouse models of primary lymphedema and that the severity of LVV defects relates to lymphedema [67]. Karkkainen et al. reported a model of primary lymphedema in two mouse strains. The Chy phenotype exhibited chylous fluid in the abdomen and foot edema. Mutagenesis using ethylnitrosourea was used to inactivate VEGFR-3. Immunohistochemistry revealed enlarged cutaneous lymphatic vessels in Chy mice, which matched the results in Milroy syndrome patients [68]. 

### 4.2. Secondary Lymphedema Animal Model

#### 4.2.1. Mouse Model

Because of their ease of use, low expense, and anatomical and histological resemblance to human lymphedema, mouse limb and tail models are frequently used in lymphatic studies [69]. The mouse forelimb lymphedema model was developed using comparable axillary lymphadenectomy techniques. The underarm lymph nodes, prenodal and postnodal collecting lymphatics, and related fat are surgically removed. By removing the axillary and brachial lymph nodes and performing partial mastectomy on the second mammary gland, Morfoisse et al. created a novel mouse model of unilateral secondary lymphedema. This forelimb model was created to be more therapeutically applicable to breast-cancer-related lymphedema than other choices by closely imitating breast cancer surgery [70]. Mendez et al. conducted axillary lymph node dissection and blocked VEGFR-3 signaling using antibody treatment. However, despite the treatment, no significant change was observed in the lymphatic outflow. These results suggest that the model used in the study may have limitations in consistently representing the chronic changes associated with secondary lymphedema [71]. 

In the studies of lymphedema in mice, the hindlimb model is more commonly used than the forelimb model. In this model, a circumferential swath of skin and subcutaneous tissue is removed above the knee, and excised lymph nodes in the popliteal, inguinal, and subiliac regions may be combined with the procedure. The prenodal and postnodal lymphatic arteries are also removed or ligated without interfering with the adjacent vasculature. However, this model may be disadvantaged because chronic lymphedema may be difficult to identify [72,73]. As a result, radiation may be required in addition to surgical excision to prolong the lymphedematous condition and better mimic human pathology [74,75]. The mouse tail model is simpler and more reproducible for studying the underlying mechanisms of lymphedema development than the hindlimb model. In this model, a 1–2 cm wide strip of tail skin is removed at a distance of 2–3 cm from the tail base, with or without maintaining the integrity of the collecting lymphatic vessels (Figure 1). This model has been widely used due to its straightforward anatomy and surgical techniques [76,77]. Another advantage of the mouse tail model is that the excision area and degree of swelling observed in this model are similar to those commonly observed in humans. This allows for investigating potential treatments for histological sequelae, such as fibrosis, fat deposition, and immune cell infiltration [78,79,80,81,82]. There are also limitations in the mouse tail model. While the mouse model has limitations in terms of its anatomical relevance to humans, another potential disadvantage is the occurrence of spontaneous lymphatic regeneration in mice. This may limit the translational relevance of findings from mouse studies to human therapeutic research.

#### 4.2.2. Rat Model

Because of its bigger anatomical size, the rat model of lymphedema provides a better visual view of its cervical, inguinal, and popliteal lymphatics when compared to the mouse model. In the rat forelimb model, the underarm lymph nodes and accompanying prenodal and postnodal collecting lymphatic arteries are removed through a long incision across the rat’s axilla. Although few studies used this model, it still provides a feasible option that represents the chronic character of lymphedema in instances of extreme fibrotic scarring [83]. In the rat hindlimb model, the lymph nodes and lymphatic arteries are removed after a circular incision in the dermal and subcutaneous regions [84,85,86]. There are also studies in rat hindlimb models combined with irradiation to promote more robust growth of lymphedema [87,88]. The surgery procedure in the rat tail model is similar to the mouse tail model. The mouse tail model is more common than the rat tail model because the rat tail model may not show as significant changes in gross metrics as the mouse tail model and has greater animal purchase and care expenses [89,90]. In addition to limb and tail models, Daneshgaran et al. developed the first reproducible rat model for head and neck lymphedema [91]. This model involves complete lymph node removal and irradiation, which results in alterations similar to human clinical postsurgical head and neck lymphedema, including delayed lymphatic drainage, subcutaneous tissue enlargement, increased fibrosis, and increased inflammation.

#### 4.2.3. Rabbit Model

The rabbit model of lymphedema offers advantages over rodent models due to its larger size and the ability to study the long-term consequences of limb swelling, in contrast to the natural resolution of lymphedema observed in rodent models. In the rabbit hindlimb model, lymphedema is induced by popliteal lymphadenectomy and lymph node and lymphatic vessel ablation, with or without irradiation [92,93]. In the rabbit ear model, lymphedema was commonly induced by circumferential excision at the ear. Its main limitation is the rabbit ear’s ability to recover rapidly, resulting in the rabbit ear scar development within three to four weeks following trauma [94,95]. 

#### 4.2.4. Large Animal Models

Large animal models such as dogs, sheep, pigs, and monkeys are valuable for anatomical and histological investigations in lymphedema research. However, these models are limited by their high cost and limited availability of molecular reagents. In the dog forelimb model, Suami et al. examined secondary lymphedema in the canine forelimb after resection of the superficial cervical and axillary nodes using the vascularized lymph node transfer [96]. This research mimics the chronic feature of lymphedema patients. In the dog hindlimb model, excision of thigh lymphatic surgery with or without radiation was used to induce lymphedema [97]. However, canine models have limitations, such as the inability to reliably generate lymphedema, the mortality rate, and the expense [96,98]. In the sheep hindlimb model, Tobbia et al. reported the development of lymphedema by popliteal lymph node excision. Baker et al. used the excision of a single popliteal lymph node and its accompanying vessels to produce lymphedema [99,100]. Although this model had a reasonably high success rate, its flaws of considerably less serious lymphatic damage than what is usually seen in real cancer patients restricted its use. Due to their size, the pig model has advantages in the detailed visualization of lymph nodes and vessel regeneration [101]. Lymphedema in the pig’s hindlimb could be produced by lymphadenectomy of the inguinal lymph nodes [101,102]. However, one of the drawbacks is the lack of axillary lymph nodes in pigs, and it is difficult to imitate axillary lymphadenectomy that caused secondary clinical lymphedema in a pig model [103]. Because of the close phylogenetic connection between monkeys and humans, the monkey lymphedema model produces more accurate results than other animal models. Wu et al. report the first reliable rhesus monkey model of upper-extremity lymphedema [104]. This model was induced by the dissection of axillary lymph nodes and lymphatic drainage vessels and the excision of subcutaneous fat and deep lymphatic tissues along with two doses of 30 Gy before and after surgery. The monkey lymphedema model has demonstrated pathologic alterations in lymphatic vessel anatomy, including the loss of main lymphatic trunks and delayed lymphatic movement. However, using primate models for lymphedema research presents significant financial and ethical concerns that must be thoroughly evaluated before selecting this model.

## 5. The Mechanism of Lymphangiogenesis

Lymphangiogenesis is the process of the formation of new lymphatic vessels from pre-existing ones. The process of lymphangiogenesis is regulated by various signaling pathways and growth factors. The main growth factor that regulates lymphangiogenesis is vascular endothelial growth factor (VEGF)-C and VEGF-D. These growth factors bind to VEGF receptor-3 (VEGFR-3) present on the lymphatic endothelial cells and induce their proliferation, migration, and tube formation. Other growth factors and cytokines such as fibroblast growth factor-2 (FGF-2), hepatocyte growth factor (HGF), transforming growth factor-β (TGF-β), platelet-derived growth factor (PDGF), and interleukin-6 (IL-6) also play a role in the regulation of lymphangiogenesis. During lymphangiogenesis, the lymphatic endothelial cells undergo several steps, including proliferation, migration, sprouting, and tube formation, which eventually results in the formation of a new lymphatic vessel. The process is also influenced by the extracellular matrix and the cellular microenvironment. Lymphangiogenesis is an important process in physiological and pathological conditions such as embryonic development, wound healing, inflammation, and cancer metastasis.

Lymphangiogenesis is induced by sprouting and elongation from existing nearby lymphatic vessels to develop mature lymphatic networks. Another mechanism of lymphangiogenesis is that circulating lymphatic endothelial progenitor cells differentiate into lymphatic endothelial cells and incorporate into new vessels. The understanding of the mechanisms of lymphangiogenesis is important in the development of therapies for lymphatic disorders and cancer treatment.

## 6. Therapeutic Lymphangiogenesis

Therapeutic lymphangiogenesis is a treatment intended to reconstruct lymphatic networks to improve lymphatic function by artificially stimulating the regeneration and reparative development of lymphatic vessels at the capillary level in an absolute or relative lymphatic drainage failure tissue (Figure 2). The network of neointimal lymphatic vessels contributes to maintaining tissue homeostasis and preventing disease progression by eliminating localized lymphatic fluid congestion and the contents, such as proteins and inflammatory cells. Lymphedema, caused by lymphatic insufficiency in the extremities, is a refractory disease with no effective fundament. Therefore, therapeutic lymphangiogenesis against lymphedema is expected to lead to novel options to recover edematous limbs by restructuring the lymphatic network at the site of the lymphatic insufficiency and consequently eliminating the lymphatic congestion. The concept of the treatment mechanism can be broadly divided into two categories: those intended to promote lymphangiogenesis by the sprouting and elongation of existing nearby lymphatic vessels and those intended to construct new lymphatic vessels by transplanting new lymphatic endothelial progenitor cells, supplying lymphatic endothelial cells, and incorporating them into endothelial cells at the site of repair. These are applications of the angiogenesis and vasculogenesis concepts in blood vessels, respectively [105]. 

### 6.1. Cell Therapy

Mesenchymal stem cells (MSCs), adipose-derived regenerative cells (ADRCs), and induced pluripotent stem (iPS) cells, which are all developed for use in multiple regenerative medicines now [106], are expected to provide a potential cell source also for therapeutic lymphangiogenesis. Conrad et al. successfully performed animal studies of lymphangiogenesis using MSC implantation to restore lymphatic drainage, indicating that MSCs play a role in lymphatic regeneration [107]. ADRCs are another promising cell source in regenerative medicine other than lymphangiogenesis. They are relatively easy to harvest from subcutaneous fat and can be easily increased by culture. In animal models, ADRCs have been reported to promote angiogenesis by secreting various pro-angiogenic factors, including VEGF-A, HGF, SDF-1, and microRNAs [108,109,110]. ADRCs could also promote lymphangiogenesis by secreting factors such as VEGF-C in ischemic tissues and improve blood perfusion by maintaining wound bed preparation in hindlimb ischemia [111]. In addition, ADRCs contribute to improving local inflammation by promoting lymphangiogenesis, resulting in cardioprotection against a high-fat-diet-induced cardiac remodeling. Importantly, ADRC implantation has been verified to improve lymphedema in a mouse lymphedema model by enhancing the secretion of several lymphangiogenic factors, such as VEGF-C and HGF, following reparative lymphangiogenesis [112]. Interestingly, the clinical study of ADRCs for breast reconstruction after mastectomy in breast cancer patients (Restore-2 trial) observed fewer onsets of secondary lymphedema in those patients with ADRC implantation, suggesting that the transplanted ADRCs may also contribute to lymphangiogenesis in clinical settings [113]. In addition, since therapeutic angiogenesis for critical limb ischemia (CLI) has already been successfully used in clinical practice and shown to be safe and effective, there are relatively few barriers to ADRCs for lymphedema in clinical use [114]. Based on the accumulating evidence and advantages, ADRCs are also an expected promising cell source for therapeutic lymphangiogenesis. The therapeutic potential of cord-blood-derived stem cells or cord-blood-derived lymphatic endothelial progenitor cells was also reported. Exosomes released from human umbilical cord mesenchymal stem cells can augment lymphangiogenesis [115], and D34+ VEGFR-3+ progenitor cells have a potential to differentiate toward lymphatic endothelial cells [116]. On the other hand, iPS cells or ES cells have been experimentally proven to differentiate into lymphatic endothelial cells, and transplantation of these cells is expected to promote lymphangiogenesis, likely with the vasculogenesis manner in terms of blood vessels [117,118,119].

### 6.2. Gene Therapy (Growth Factors)

Translational studies of lymphangiogenesis using gene therapy have long involved strategies that induce strong expression of major lymphangiogenic factors such as VEGF-C [120,121] and VEGF-D [122]. Other strategies in an animal model have demonstrated the efficacy of therapeutic lymphangiogenesis against lymphedema to increase or supplement the growth factors or peptides such as HGF [123], Adrenomedullin [124], bFGF [125], and Angiopoetin-1 [126]. On the other hand, therapeutic lymphangiogenesis using [127] or TGFβ1 [128] has been unsuccessfully tested. The question of why these factors have negative results in lymphedema models regardless of the potential to act on lymphatic endothelial cells and enhance pro-lymphangiogenic activity at the in vitro level should be examined to improve and apply gene- or growth-factor-based therapeutic lymphangiogenesis in the future. Currently, HGF is being developed for clinical application in therapeutic angiogenesis for CLI, and the success of this trial is expected to have fewer barriers to its clinical application against lymphedema.

### 6.3. Proteins, Peptides, miRNA, Drugs

Energetic studies for exploring therapeutic lymphangiogenesis for lymphedema by the administration of proteins or other therapeutic applications targeting specific molecules are also under investigation [129]. Adiponectin, one of the cytoprotective adipokines [76], hydrogen sulfide (H_2_S), which is emerging as one of the gas transmitters [77], and exosomes [130] have been attracting attention as they can act on lymphatic endothelial cells and enhance lymphangiogenesis. In addition, their effects on improving lymphedema have been demonstrated in animal models. Cilostazol in a protein kinase A-dependent manner [131], 9-cis retinoic acid with the regulation of p27^Kip1^, p57^Kip2^, and aurora kinases [132,133], hyaluronidase through the degradation of HA trafficking [134], COX-2 via the generation of prostaglandins (PGs) [135], ketoprofen through enhanced VEGF-C signaling [136], Leukotriene B4 antagonism modulating Notch signaling [137], and ketone body oral intake with the recovery of OXCT1 levels [138] have also been shown to promote lymphangiogenesis. Those are all candidates to modulate pathological lymphatics to treat lymphedema. However, it is important to consider the route of administration, how to maintain the effective pharmacological concentration, and the proper frequency of the administration of these drugs to determine whether they are effective in the actual clinical setting by preclinical studies.

### 6.4. Others

Other techniques for therapeutic lymphangiogenesis have also been reported. For example, LN transfer is one of the most well-studied methods. The mechanism of LN transfer in lymphedema was reported to be the increased production of pro-lymphangiogenic growth factor (i.e., VEGF-C) and antifibrotic cytokine (i.e., IL-10). It has been demonstrated as a novel strategy in clinical application [139,140,141,142,143,144]. Low-energy shock wave therapy [90] and low-level laser therapy [145] are other interesting therapeutic strategies. The underlying mechanisms were indicated to induce therapeutic lymphangiogenesis by up-regulating VEGF-C and bFGF [90]. 

## 7. Clinical Study

There are some clinical trials regarding therapeutic lymphangiogenesis (Table 1). Adenoviral VEGF-C overexpression combined with LN transfer was conducted in a Phase 1 trial to evaluate its short-term safety [146]. The study demonstrated a 36-month efficacy and 5-year safety follow-up of the patients. HGF treatment is an ongoing study; its results are not available yet (UMIN000033159). In turn, as cell therapy, BM-stromal cells were used for fifteen patients and were shown to be effective and feasible in limited cases [147]. A study on BM-MNCs reported that injecting 10 women with ASC in the affected arm reduced the volume of lymphedema compared to the control group which comprised 10 women and received traditional compression sleeve therapy (CST) [148]. A case report and another pilot study have shown that ADRC implantation for secondary lymphedema can improve daily symptoms and reduce the volume of the affected arm while being considered safe [149,150,151]. However, these clinical reports are still initial pilot studies with limited cases in Phase 1 trials. A Phase II study has been initiated to test further its feasibility and safety. Lymfactin (AdAptVEGF-C adenoviral vector) is a promising therapeutic approach currently undergoing clinical trials (NCT03658967).

## 8. Limitation and Future Perspective

Previous studies in animal models have primarily focused on examining the efficacy of therapeutic lymphangiogenesis in improving lymphedema during the acute inflammation phase that occurs just after lymphangectomy. However, inflammation during this phase could induce active tissue regeneration and endogenous lymphangiogenesis. This means that successful therapeutic lymphangiogenesis in this model may not necessarily translate to effectiveness during the chronic phase of the disease when inflammation has subsided. Moreover, clinical lymphedema typically develops several years after lymph node dissection surgery, which raises important considerations such as the timing of treatment and the route of administration to the site of lymphatic insufficiency. There is also a need for small animal models that can mimic the chronic condition of clinical lymphedema to further advance research in this field. Despite these challenges, therapeutic lymphangiogenesis holds promise as a fundamental treatment for patients with refractory lymphedema.

## 9. Conclusions

Therapeutic lymphangiogenesis is a promising novel therapeutic strategy to reconstruct the fragmented lymphatic network and restore lymphatic function against refractory lymphedema. 

## Figures and Tables

**Figure 1 ijms-24-07774-f001:**
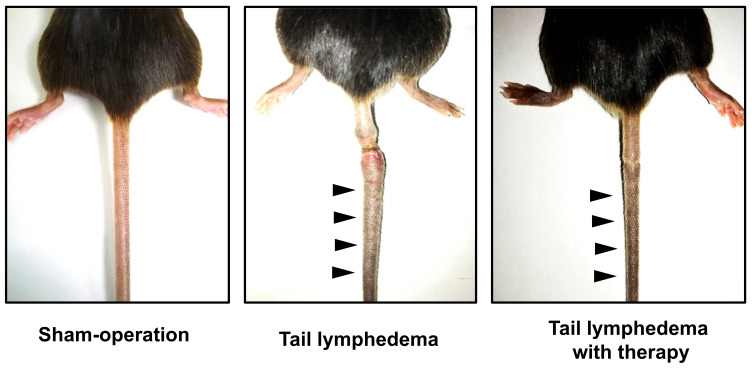
Mouse tail lymphedema model: **left** panel: sham; **middle** panel: lymphedema; **right** panel: lymphedema with therapeutic lymphangiogenesis by cell therapy. Black arrowheads indicate tail swelling.

**Figure 2 ijms-24-07774-f002:**
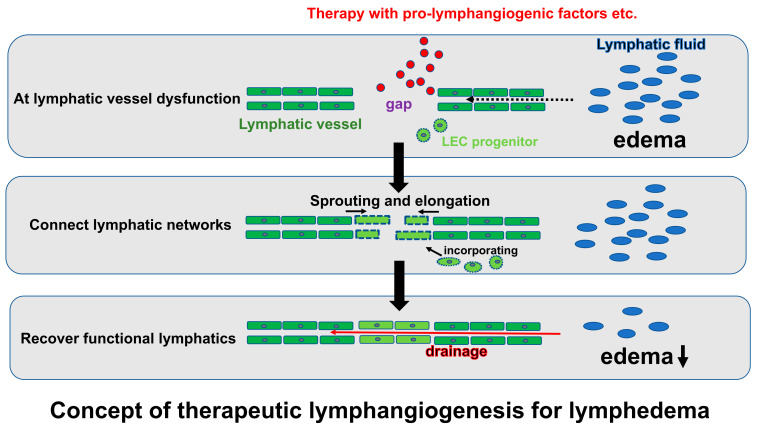
Schema for the concept of therapeutic lymphangiogenesis against lymphedema. LEC indicates lymphatic endothelial cell.

**Table 1 ijms-24-07774-t001:** Clinical trials of therapeutic lymphangiogenesis for lymphedema.

	Target Molecule	Patient Number	Edema Site	Follow-Up	Result	Year	Ref.
Gene therapy	VEGF-C	15	Upper limb	12 months	No dose-limiting toxicities and well tolerated.	2020	Hartiala et al. [146]
Cell therapy	BM-stromal cell	15	Upper limb	12 months	The BMSC Group had a reduction in lymphedema volume and pain scale and a better long-term cure result.	2008	Hou et al. [147]
	BM-MNC	10	Upper limb	3 months	BM-MNCs reduce lymphedema volume and chronic pain and improve sensitivity.	2011	Maldonado et al. [148]
	ADSC	1	Upper limb	1 and 4 months	Symptoms in patients were improved over time, and volume of affected arm was reduced.	2016	Toyserkani et al. [149]
	ADSC	10	Upper limb	1, 3, and 6 months	Non-significant change in volume was observed. Patient outcomes improved significantly over time. Half of the patients reduced their use of conservative management.	2017	Toyserkani et al. [150]
	ADSC	10	Upper limb	1, 3, 6, and 12 months	No significant change in volume was observed. Patient outcomes improved significantly over time. Half of the patients reduced their use of conservative management.	2019	Toyserkani et al. [151]
Others	LN transfer	20	Upper limb	39 months	Patients exhibit decrease in cellulitis, circumferential reduction, and circumferential differentiation.	2013	Cheng et al. [152]
	LN transfer	15	Limb	12 months	Mean episodes of cellulitis and circumferential difference and the overall lymphedema quality of life were improved.	2018	Asuncion et al. [153]

BM-MNC indicates bone-marrow-derived mono nuclear cell. ADSC indicates adipose-derived stem cell. LN indicates lymph node.

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
