# Peer review of "Therapeutic Lymphangiogenesis Is a Promising Strategy for Secondary Lymphedema"

_ijms, 2023, doi:10.3390/ijms24097774_

Round 1
Reviewer 1 Report
Analysis is well conducted and many data are present in this review. Nevertheless, I have some suggestion:
1. The manuscript needs a thorough revision in order to correct English language mistakes. I had started to write down some mistakes I noticed, but there were so many, that I have not listed all. For example,
1) The terms “In contrast” is not very suitable in Line 55 and Line 108.
2) Line 105: There is an absence of period after the “[18]”.
3) And please rephrase the sentence Line 105-108 “Infections caused for secondary lymphedema include filariasis, cellulitis/erysipelas, tuberculous lymphadenitis, and lymphogranuloma venereum, which filariasis is the most common cause of secondary lymphedema worldwide.”
2. Line 85-101, Line 123-130, Line 151-163, and Line 282-295: It seems that “primary lymphedema” is not strict necessary for the proof of the main thesis.
3. Instead, it would be beneficial if “The mechanism of lymphangiogenesis” was described in a single section before “4. Therapeutic lymphangiogenesis (Line 397)”. And more details about lymphatics hierarchical networks needs to provide (Line 48-72)
4. Line 511-512: Maybe, we’d better added citation number for each clinical trials in Table 1
5. The citations that within the last three years was merely 15.5% (23/148).
Reviewer 2 Report
line 122
In epidemiology part:
Only mentioned Statistics of America and United Kingdom to secondary lymphedema
Authors need to add the Statistics of Asia countries studies
line 178
Treatment :
Explain the limitation of each treatment
Line 276
Animal lymphedema model:
Authors need to detail explain of the animal experimental method and results of (Figure 1)
Lin2 418
Cell therapy
Please add the therapeutic potential of cord blood derived stem cell or cord blood derived lymphatic endothelial progenitor cells.
Line 397
Therapeutic lymphangiogenesis.
Authors mentioned two categories for Therapeutic lymphangiogenesis. However only one category was described in figure 2. Correct the figure2 to explain two categories of Therapeutic lymphangiogenesis
